# Identification and Functional Analysis of NLP-Encoding Genes from the Postharvest Pathogen *Penicillium expansum*

**DOI:** 10.3390/microorganisms7060175

**Published:** 2019-06-15

**Authors:** Elena Levin, Ginat Raphael, Jing Ma, Ana-Rosa Ballester, Oleg Feygenberg, John Norelli, Radi Aly, Luis Gonzalez-Candelas, Michael Wisniewski, Samir Droby

**Affiliations:** 1Department of Postharvest Science, Agricultural Research Organization, the Volcani Center, Rishon LeZion 7505101, Israel; elena.levin@gmail.com (E.L.); pongie@volcani.agri.gov.il (G.R.); fgboleg@volcani.agri.gov.il (O.F.); 2Appalachian Fruit Research Station, Agricultural Research Service, United States Department of Agriculture, Wiltshire Road, Kearneysville, WV 25430, USA; Jing.ma@ars.usda.gov (J.M.); jnorelli@afrs.ars.usda.gov (J.N.); michael.wisniewski@ars.usda.gov (M.W.); 3Instituto de Agroquímica y Tecnología de Alimentos (IATA-CSIC), Calle Catedrático Agustin Escardino 7, Paterna 46980, Valencia 46980, Spain; ballesterar@iata.csic.es (A.-R.B.); lgonzalez@iata.csic.es (L.G.-C.); 4Department of Plant Pathology and Weed Research, ARO, the Volcani Center, Newe-Yaar Research Center, Ramat Yeshai 30095, Israel; radi@volcani.agri.gov.il

**Keywords:** blue mold, apple, Nep1, virulence, pathogenicity, effectors

## Abstract

*Penicillium expansum* is a major postharvest pathogen that infects different fruits, mainly through injuries inflicted during harvest or subsequent handling after harvest. Several effectors were suggested to mediate pathogenicity of *P. expansum* in fruit tissue. Among these effectors Nep1-like proteins (NLPs), produced by various microorganisms with different lifestyles, are known for their ability to induce necrosis in dicot plants and were shown to be involved in virulence of several plant-related pathogens. This study was aimed at the identification and functional characterization of two NLP genes found in the genome of *P. expansum*. The genes were designated *Penlp1* and *Penlp2* and were found to code type1 and type3 NLP respectively. Necrosis-inducing activity of the two proteins was demonstrated by transient expression in *Nicotiana benthamiana* leaves. While *Penlp1* expression was induced during apple infection and in liquid culture, the highest level of *Penlp2* expression was found in ungerminated spores. Deletion of *Penlp1*, but not *Penlp2*, resulted in reduced virulence on apples manifested by reduced rate of lesion development (disease severity).

## 1. Introduction

Necrotrophic fungal pathogens of plants promote host cell death through the secretion of toxins and acquire nutrients for growth from dead cells [1]. In the early stages of infection and disease development, a wide-range of virulence strategies are employed by these pathogens to overcome the innate immune system of their hosts. These include the secretion of diverse phytotoxic compounds, cell-degrading enzymes, and proteinaceous effectors that are involved in manipulating host defiance mechanisms [2]. Only a relatively few protein effectors in necrotrophic pathogens, however, have been described and the molecular interactions between these effectors and host plants, in most cases, remain to be elucidated.

*Penicillium* is a wide-spread fungal genus with 354 recognized species [3] that are distributed globally, where they inhabit a large variety of substrates, from soil to foods. *Penicillium expansum* is a postharvest necrotrophic pathogen that infects a wide range of fruits, including pome and various stone fruit [4,5]. Infection of fruit occurs through peel injuries where nutrients stimulate spore germination. Several reports on the potential role of different factors that mediate pathogenicity and virulence in *P. expansum* on apple fruit have been published in recent years. Acidification of host tissue leading to enhanced production of plant cell wall degrading fungal enzymes, as well as production of patulin are among the suggested factors influencing the pathogenicity and virulence of *P. expansum* [6,7,8,9]. In a study reported by Sánchez-Torres, the *PeSte12* transcription factor was shown to play a role in virulence and that knocking out this gene in *P. expansum* significantly affected the rate of decay development on apple fruit stored at 0 °C. No significant effect, however, was observed when apple fruit was inoculated with knockout mutants and stored at 20 °C [10].

In our previous works [11,12], the potential role of *PeLysM* and subtilisin-related peptidase *Peprt* genes in the pathogenicity and virulence of *P. expansum* on apples was investigated. Deletion of *Peprt* in *P. expansum* resulted in reduced sporulation and virulence of the fungus on apples [12]. Expression level of four *LysM* coding genes was induced during the infection of apple fruit and decay development. No significant effect on decay incidence and severity was observed however, when each of these genes was knocked out separately. Nevertheless, a wide range of apple proteins, including those involved in plant defense processes and iron sequestration, were demonstrated to interact with fungal LysM proteins using a yeast two-hybrid system. This suggested that LysM effector proteins in *P. expansum* may have a potential role in modifying host response to the pathogen during the infection process.

Effector proteins are generally species-specific and lack common features within and across species [13]. Several effector families, however, have been demonstrated to be secreted by various plant pathogens. Among the most studied fungal effectors are the necrosis and ethylene-inducing peptide 1–like proteins (NLPs). These have been identified in bacteria, fungi and oomycetes [14,15,16]. The first member of this protein family was identified by Bailey et al [17] in culture filtrate of *Fusarium oxysporum* and was demonstrated to induce ethylene production and necrosis in coca leaves *(Erythroxylum coca).* Analysis of more than 500 NLP sequences from fungi, oomycetes, and bacteria identified four different groups belonging to three phylogenetic clusters [16]. Type 1 NLPs are found mostly in plant-associated bacteria and carry all the features that were shown to be essential for cytotoxic activity. A large number of Type 1a NLP was found in oomycetes. Type 1a diverged from type 1, but lacks the ability to cause necrosis, probably due to amino acid substitutions that have occurred in the acidic cation-binding pocket [16]. Distinct from type 1, type 2 NLPs possess a second conserved disulfide bridge and a putative calcium-binding domain. Type 3 NLPs are different from the other types in the amino acid sequence of N- and C-terminal portion, and there is still very little experimental data available regarding type 3 NLP [16].

The most known feature of NLPs is their ability to induce necrosis and ethylene production in leaves of dicot, plant species [14,16]. However, there are also many non-cytotoxic NLPs that have been described in the literature. Among the tested NLPs, two of three in *Phytophthora infestans* [18], 11 of 19 in *Phytophthora sojae* [19], all 12 in *Hyaloperonospora arabidopsidis* [20], five of seven in *Verticillium dahliae V592* [21], and one of two in *Colletotrichum higginsianum* [22] do not induce necrosis when transiently expressed in plants. Likewise in *Phytophtora capsici*, only four out of 39 NLP-coding genes were demonstrated to have cytotoxic activity [23] and only two of the seven NLPs in *V. dahlia VdLs* induce plant cell death [24]. Ten different non-cytotoxic NLPs in the biotrophic, downy mildew pathogen, *H. arabidopsidis* do not induce necrosis, but strongly activate the plant immune system in *Arabidopsis thaliana* [25]. Oome and coworkers [25] reported that highly-conserved peptide from non-cytotoxic, as well cytotoxic type 1 NLPs, are able to induce the immune response in *Arabidopsis.* Bohm and coworkers demonstrated that although some NLPs (such as PpNLP from *Phytophthora parasitica*) able to trigger pattern-induced immune response, others (such as PccNLP from *Pectobacterium carotovorum*) depend on the cytotoxic activity in order to induce immune response [26]. The conserved peptide that is crucial to trigger immune response was pinpointed [25,26], however, its ability induce immune response was shown to be restricted only to certain plant families [26]. Likewise, a functional diversification exists in the superfamily of NLP1-like proteins with some members contributing to virulence while others do not. Notably, deletion of NLP1, one of the two cytotoxic NLP in *V. dahlia VdLs* affected vegetative growth and conidiospore production. Moreover, deletion of either *NLP1* or *NLP2* compromised *V.dahlia VdLs* virulence [24]. Additional NLPs have been reported to contribute to the virulence of several pathogenic microorganisms. Deletion of NLPPcc of the soft-rot bacterium *Erwinia (Pectobacterium) carotovora* subsp. *carotovora* reduced its virulence on potato tubers [27]. Dallal-Bashi et al. [28] reported that transient expression of SsNep1 and SsNep2 NLPs from *Sclerotinia sclerotiorum*, in tobacco leaves induced necrosis formation, while reduction in *SsNep2* expression resulted in a partial or total loss of virulence [28]. BcNEP1 and BcNEP2, two paralogous of NLP from *Botrytis cinerea*, caused necrosis in dicotyledonous plant species, however, their role in *B. cinerea* virulence is still unclear [29]. Loss of a single NLP gene in organisms containing multiple NLPs, in some cases does not affect virulence [21,24,30,31] possibly due to genetic redundancy. Moreover, deletion of the single NLP gene of the wheat pathogen *Mycosphaerella graminicola* did not affect its virulence [32] suggesting that in some organisms, the role of NLPs in the infection process is not crucial.

In the present study, the identification and functional characterization of two NLP genes found in the genome of *P. expansum* is presented. Both genes encode putative, secreted cysteine-rich proteins. PeNLP1 and PeNLP2 are type 1 and type 3 NLPs, respectively. *Penlp1* was found to be strongly expressed during vegetative growth and in liquid culture, while *Penlp2* was expressed in spores. Both PeNLP1 and PeNLP2 are phytotoxic when they are transiently produced in *Nicotiana benthamiana* leaves. Loss of PeNLP1, but not PeNLP2, appears to affect *P. expansum* virulence on apple fruit.

## 2. Materials and Methods

### 2.1. Plant Material

Apple fruits (var. Golden Delicious) at harvest maturity were collected from harvest bins in an apple packing house ("Fruit Summits") that belongs to the Agricultural Cooperation Association (ACA) in northern Israel. Fruit were used in experiments either shortly after harvest or kept in a cold storage room at 0 °C for later use.

### 2.2. Fungal Cultures

*P. expansum* strain PEX2 (MD-8, Accession: PRJNA255747 ID:255747) isolated in the USA from infected apple fruit, was used in the current and previous studies [11,12,33]. *P. expansum* cultures were grown on PDA plates at 25 °C for 2–3 weeks. Harvesting spores from fungal cultures was done using sterile bacteriological loop and suspending them in sterile distilled water. To prepare spore suspensions at the desired concentration, spores were pelleted by centrifugation at 10,000 rpm for 5 min, resuspended in sterile distilled water and final concentration was adjusted using the appropriate dilution. For liquid culture, 1 mL of 10^6^ spores/mL was inoculated into 100 mL of a PDB medium and incubated at 25 °C in still cultures.

### 2.3. Phylogenetic Tree Construction

The alignment of NLP protein sequences was done using software: Molecular Evolutionary Genetics Analysis version 7.0 (MEGA7, Oxford University, England, UK) [34]. The evolutionary history was done using the Neighbor-Joining method [35]. The evolutionary history tree showing the sum of branch length = 5.72043509 is presented in Figure 1. Phylogenetic tree was constructed using branch lengths in the same units as those of the evolutionary distances. The percentage of replicate trees in which the associated taxa clustered together in the bootstrap test (100 replicates) are shown next to the relevant branches. The computation of the evolutionary distances was done with Poisson correction method [36] and are presented in units of the number of amino acid substitutions per site. To perform the analysis, 19 amino acid sequences were involved, and all gaps and missing data were eliminated from the appropriate positions. There was a total of 180 positions in the final dataset. Evolutionary analyses were conducted in MEGA7.

### 2.4. Disulfide-Bond Prediction

X-ray structures of NEP-1 from *Moniliophthora perniciosa* (PDB ID 3ST1) [37], and a 25kDa protein elicitor from *Pythium aphanidermatum* (PDB ID 3GNU, 3GNZ) [38] were used as templates in the attempt to build the 3D structure model of PeNLP1. Six different methods were used to model the target’s structure: Swiss-model [39], RaptorX [40] HHPRED [41], FFAS03 [42], Phyre2 [43] and i-Tasser [44]. Qmean [45], MolProbity [46], Verify3D [47] and ProQ2 [48] were used for quality assessment and selection of the best model structure.

### 2.5. Transient Expression in Nicotiana Benthamiana Leaves

Plasmid construction: *Penlp1* and *Penlp2* open reading frames with or without signal peptide were amplified from cDNA using primer pairs: 41/42, 43/42, 44/45 and 46/45 (Appendix A). PCR products were cloned into pRTL2 [49] plasmid using *BamH*I and *Xba*I sites constructing pRTL2_NLP1_SP, pRTL2_NLP1_NS, pRTL2_NLP2_SP and pRTL2_NLP2_NS plasmids. Cassettes containing the 35S transcriptional promoter originated from cauliflower mosaic virus, the tobacco etch virus 5’ nontranslated region sequence, the *Penlp1/Penlp2* open reading frame and 35S transcriptional terminator were removed from these plasmids using *Hind*III restriction enzyme and the fragments were subsequently cloned into the binary plant transformation vector pBINPLUS/ARS [50] resulting in pBINPLUS/ARS_NLP1_SP, pBINPLUS/ARS_NLP1_NS, pBINPLUS/ARS_NLP2_SP, and pBINPLUS/ARS_NLP2_NS, respectively. The plasmids were transformed into *Agrobacterium tumefaciens* GV3101 using electroporation.

Infiltration procedure: Transient expression in *N. benthamiana* leaves was accomplished using the protocol of Sparkes et al. [51] with slight changes. Briefly, Agrobacterium containing the binary vectors was grown for approximately 15 h in yeast extract beef (YEB) broth (1 g/L yeast extract, 5 g/L beef extract, 5 g/L peptone, 5 g/L sucrose, 0.5 g/L MgSO_4·_7H_2_O) with 30 µg/mL and 50 µg/mL of gentamicin and kanamycin, respectively, at 28 °C, 200 rpm. Agrobacterium cells were pelleted using centrifuge at 2500 *g* for 10 min at room temperature, washed by infiltration medium (5 g/L d-glucose, 50 mM MES, 2 mM Na_3_PO_4_, 0.1 mM acetosyringone), and centrifuged again. The pelleted cells were then diluted in infiltration medium to OD_600_ = 0.1. After incubation for 1.5 h, a cell suspension was infiltrated into leaves of two-week old *N. benthamiana* plants using a 1 mL syringe without a needle.

Detection of β-glucuronidase (GUS): One cm diameter leaf-discs were removed with a cork borer from the infiltration site, placed in a 24-well culture plate, with each well containing 1 mL fresh staining buffer (0.1M sodium phosphate pH 7, 10 mM EDTA pH 8, 0.1% *v*/*v* Triton X-100, 1 mM K_3_Fe(CN)_6_, 2 mM X-Gluc (5-bromo-4-chloro-3-indolyl-β-d-glucuronic acid) following incubation for 15 h at 37 °C.

### 2.6. Fruit Wounding, Inoculation, and Disease Assessment

Before use, apple fruit were surface disinfected by dipping in a 2.5% commercial bleach solution for 5 min, rinsed with tap water and then kept at room temperature to dry. Fruit were wounded at four sites around the stem end of the fruit using a needle to make a 3 mm deep wound, and each wound was inoculated with 10 µL of a spore suspension of *P. expansum* at 10^5^ spores/mL Inoculated fruit were then placed in plastic boxes covered with polyethylene sheets and incubated at 25 °C. Each treatment consisted of fifteen fruit and each fruit served as one replicate. Disease incidence and severity (lesion dimeter) was examined at 3, 4, 5, and 6 dpi.

### 2.7. RNA Extraction and cDNA Synthesis

Total RNA was extracted from *P. expansum*-infected apple fruit tissue at different times starting at 24 h post inoculation (hpi) using the protocol described by Levin et al. [11,12]. The mycelia from a liquid culture of *P. expansum* was collected by filtration through Whatman filter paper, immediately frozen in liquid nitrogen and then freeze-dried. RNA extraction was done using the SV Total RNA Isolation System (Promega, Madison, Wisconsin, USA). Quality of total RNA was analyzed by gel electrophoresis and the concentration was measured using spectrophotometer ND-1000 (NanoDrop, Wilmington, DE, USA). Verso cDNA Kit (Thermo Fisher Scientific, Epson, UK) was used to synthesize first-strand cDNA from 1 μg of total RNA pretreated with 0.25 units of recombinant DNase I (Takara Bio Inc., Shiga, Japan).

### 2.8. Reverse Transcription-Quantitative PCR (RT-qPCR) and RNA-Seq Analysis

PCR was done using a Labcycler (Senso Quest, Göttingen, Germany) thermocycler. Each PCR sample contained 1U DreamTaq DNA polymerase (Thermo Scientific, Waltham, MA, USA), DreamTaq buffer, 0.2mM DNTPs, 0.4 µM of each primer as detailed below, 1ng–1µg template DNA (in total volume of 10 µL). The cycling conditions were: 5 min at 94 °C, following 35–40 cycles of: 30 s at 94 °C, 15 s at 54-58 °C, 1–3 min at 72 °C; and 10 min at 72 °C. Primer sequences are listed in Appendix A.

RT-qPCR was performed with a StepOnePlus™ Real-Time PCR System (Applied Biosystems, Foster City, CA, USA). RT-qPCR reaction mixtures contained 3 μL of cDNA template in 10 μL of a solution containing 5 μL of Fast SYBR® Green Master Mix (Applied Biosystems, Foster City, CA, USA) and 300 nM primers as detailed below. Cycling conditions were 20 s at 95 °C, followed by 40 cycles of 3 s at 95 °C and 30 s at 60 °C.

Results were analyzed using LinRegPCR version 2014.8 [52]. The 28S rRNA gene, 37S rRNA gene, s24 and Histone H3 served as endogenous controls, and a spore sample served as a calibration sample. Relative quantification was calculated using the mathematical model of Pfaffl [53]. Each analysis comprised three different biological replicates and the experiment was repeated three times.

### 2.9. Construction and Analysis of Knockout Mutants

*Penlp1* and *Penlp2* knock out mutants were constructed using the protocol described previously by Levin et al. [11] with slight modifications. The regions of around 1500 bp 5’ upstream (promoter) and 3’ downstream (terminator) of PEX2_080220 and PEX2_071150 genes, were PCR amplified from genomic DNA of *P. expansum* PEX2, using 19/20, 21/22, 23/24, 25/26 primer pairs (Appendix A), respectively. The cycling conditions were: 4 min at 94 °C, following 40 cycles of 30 s at 94 °C, 15 s at 56 °C, and 2 min at 72 °C, and a final elongation step of 10 min at 72 °C. Analysis of transformants to ensure proper gene disruption was done by PCR. Appendix A shows schematic representation of the corresponding wild-type and gene-disrupted loci. The insertion of the selection marker was checked with primer pair 3/4 (Appendix A; Appendix A). The testing of deletion of the target genes *Penpl1* and *Penlp2* genes was checked by 1/2 and 9/10 primer pairs respectively. Integration of the T-DNA by homologous recombination was examined with primer pairs 5/6 and 7/8 for *Penlp1* and with 11/6 and 7/12 for *Penlp2* (Appendix A). Determination of the number of T-DNA copies that had been integrated in the genome of the transformants was performed using real-time genomic PCR analysis, following the protocol described by Crespo-Sempere and associates [54], using PEX2 as control. Two primer pairs, 13/14, and 15/16 were designed within the T-DNA in the promoter regions of the target genes, close to the selection marker. The *P. expansum* β-tubulin gene (AY674401) [55] was used as reference gene using the primer pair 17/18 (Appendix A). The expression of the target genes during apple infection was done by RT-PCR using NLP1 complete F/R and NLP2 complete F/R primer pairs (Appendix A).

### 2.10. P. expansum Radial Growth, Sporulation and Germination Assays

Radial growth was examined by introduction a 5 μL of a spore suspension (10^6^ spores/mL) in the center of a PDA plate and incubating it at 25 °C. Conidia production was quantified as follows: 5 mL of water was added to an 11 day-old *P. expansum* PDA culture and spore suspensions were prepared by robbing the plate with sterile spreader. After collecting the suspension, each plate was washed with additional 2 mL sterile water to collect remaining spores. Fifty fold diluted spore suspensions were counted in a hemocytometer. Spore germination was evaluated by calculation of percent germination and germ-tube length using NIS-Elements BR Microscope Imaging Software (Nikon Instruments Inc., San Francisco, CA, USA). Microscope images of the germinating spores were taken following 18 h incubation at 25 °C of 15 μL spore suspension (2 × 10^5^ spores/mL) inoculated on 15% agar-water plates.

## 3. Results

### 3.1. Genes Coding for NEP-1 Like Proteins in the Genome of P. expansum

In our previous work, a bioinformatic pipeline designed for effector-prediction in *P. expansum,* identified a NLP coding gene, PEX2_080220, a gene encoding a protein that was potentially involved in the *P. expansum* pathogenicity and virulence on apple [12]. Further analysis of the annotated, complete genome sequence of *P. expansum* (ASM76974v1) using blastn or blastp algorithms did not identify any other sequences that were homologous to PEX2_080220. Screening the annotation of the genome, however, revealed an additional gene (PEX2_071150) possessing a NPP-1 domain (IPR008701). Based on this criteria, it appears that PEX2_080220 (denoted as *Penlp1*) and PEX2_071150 (denoted as *Penlp2*) represent the entire number of NLP genes in the *P. expansum* PEX2 genome.

### 3.2. Phylogenetic Analysis and Prediction of Disulfide Bonds in PeNLP1 and PeNLP2

To determine the relationship of PeNLP1 and PeNLP2 to the reported NLP subfamilies, a phylogenetic analysis of PeNLPs was conducted together with seven representatives of type 1 fungal NLPs, five fungal type 2 NLPs, and five fungal type 3 NLPs. The analysis resulted in a neighbor-joining tree that consisted of three groups of NLPs (Figure 1A), where PeNLP1 is present in the type 1 group and PeNLP2 is in the type 3 group. Both proteins possess a predicted N-terminal signal peptide, suggesting that they are secreted. PeNLP1 and PeNLP2 possess 5 and 6 cysteines, respectively (Figure 1B). Both sequences were aligned with the conserved 24-aa peptide (nlp24) that was demonstrated to be sufficient to trigger immune response in plants [25,26]. The first part of the conserved region (AIMYAWYFPKD), which is likely to contribute to the induction of plant immune response [25] is conserved in PeNLP1, besides the substitution of two amino acids: isoleucine to leucine and alanine to serine (Figure 1C). In PeNLP2, however, this region is mostly degenerated excluding conserved tyrosine (Y), lysine (K) and aspartic acid (D) (Figure 1C). PeNLP1 has an intact heptapeptide “GHRHDWE” motif, while the third amino acid of this motif in PeNLP2 is changed from Arg to Asp, and the fourth amino acid is changed from His into Tyr (Figure 1C).

Most of the known type 1 and type 2 NLPs contain two conserved cysteine residues [25]. In the 3D structures of NLP available in the literature, these cysteines were shown to form a disulfide bond [25]. PeNLP1, however, has an additional three cysteine residues. To determine the potential of additional disulfide bonds, the 3D structure of PeNLP1 was modeled using the x-ray structures of type 1 NLP from *Moniliophthora perniciosa* (PDB ID 3ST1) [37] and a 25kDa protein from *Pythium aphanidermatum* (PDB ID 3GNU, 3GNZ) [38] as a template reference. Six different methods were used to model the 3D structure of PeNLP1. The best model structure was built using Swiss-model, with 3ST1 as template (Figure 2A). According to this model structure of PeNLP1, there is a disulfide bond between Cys64-Cys90, corresponding to the conserved disulfide bond of type 1 NLPs. In the constructed model, another two cysteines (Cys153 and Cys162) are located in close proximity to each other (Figure 2B), although the constructed model does not predict disulfide bond formation between these two cysteines. This does not, however, eliminate the potential formation of an additional disulfide bridge in PeNLP1 since the coordinates of the constructed model are biased towards the template structure, which does not contain any cysteine residues in the positions that match residues CYS153 and CYS162.

### 3.3. Cytotoxicity of PeNLP1 and PeNLP2 in Tobacco Leaves

Twelve of the fourteen tested Type 1 NLPs have been reported to induce necrosis [16]. In the present study, a transient expression system in tobacco (*N. benthamiana*) was used to evaluate the necrosis-inducing activity of PeNLP1 and PeNLP2. The ORF of each gene with and without the inclusion of the endogenous signal peptide sequence was cloned under the control of the 35S transcriptional promoter from cauliflower mosaic virus, inserted into a pBINPLUS/ARS vector [50] and transformed into *A. tumefaciens* GV3101.

The transgenic *A. tumefaciens* strains were subsequently infiltrated into the leaves of tobacco plants. Chlorotic spots appeared around the infiltration site at 3–5 days after the agroinfiltration of PeNLP1 (Figure 3). Necrosis of the tissue around the infiltrated site was apparent 10 days after agroinfiltration of PeNLP1 constructs with a signal peptide (Figure 3). Infiltration of constructs without signal peptide resulted only in minor chlorosis and several minor necrotic spots (Figure 3A). Similarly, leaves infiltrated with constructs coding for PeNLP2 with signal peptide developed stronger necrosis than leaves infiltrated with PeNLP2 constructs without a signal peptide. The necrosis induced by PeNLP2, however, was more restricted, relative to the necrosis induced by PeNLP1, and developed 12 days after the infiltration. No signs of necrosis or other visible damage was observed in leaves in response to agroinfiltration with the GUS gene (Figure 3). GUS was chosen as a negative control, since it does not have any cytotoxic activity [16] and its expression can be easily verified and visualized (Figure 3B).

### 3.4. In Planta and in Vitro Expression of Penlp1 and Penlp2

*P. expansum* spores were inoculated into artificial wounds on ‘Golden Delicious’ apple fruit and infected tissue was subsequently collected 24, 48, 72, and 96 h post inoculation (hpi). RT-qPCR analysis was conducted on the collected samples to assess *Penlp1* and *Penlp2* transcript levels using the transcript levels of three reference genes (28S rRNA, 37S ribosomal protein s24 PEXP_041240, and histone H3 PEXP_016920) for normalization. Results indicated that *Penlp1* was strongly induced, reaching a maximum expression level at approximately 24 hpi, which remained constant for up to 96 h (Figure 4A). A similar expression pattern was observed from the analysis of the RNA-Seq experiment using a mixture of RNAs from healthy apples and fungal spores (denoted as time 0) and also apple fruits inoculated with *P. expansum* isolate PEX1 (CMP-1, Accession: PRJNA255744 ID:255744, SRA accession number SRP043407) after 24, 48 and 72 h post inoculation (Figure 4B) [33]. In contrast to *Penlp1*, the expression of *Penlp2* was relatively high in spores but decreased approximately five-fold during infection and decay development (Figure 4A). No evidence of *Penlp2* expression was found in the transcriptome of PEX1 during the infection of apple fruits [33].

Interestingly, when the full-length ORF of *Penlp2* was amplified by PCR using cDNA from spores and *P. expansum* mycelium grown in culture as a template, three bands were observed (Appendix A). The largest band in spores was the main product with the addition of two, smaller, fainter bands. In mycelium sampled at 72 and 96 h of growth in culture, however, the two smaller, fainter products were now stronger and more evident. Each of the three bands was extracted from the gel and cloned into pGEM vector. Sequence analysis of the clones revealed three splice variants: variant 1 (var1) –993 bp with no introns, var2 –891 bp with one 102 bp intron, and var3 –879 bp with two introns, which is similar to the prediction in the *P. expansum* PEX2 genome (Appendix A). While two introns in the var3 are canonical GU-AG introns that carry conserved fungal 5’ donor and 3’ acceptor sites [56], the alternative intron in var2 is not a canonical intron. Moreover, translation of var1 and var2 resulted in a truncated protein due to the introduction of in-frame stop codon.

Construction of *Penlp1* and *Penlp2* knockout mutants of strain PEX2 was applied to study the potential role of PeNLP1 and PeNLP2 in pathogenicity of *P. expansum* on apples. The knockout mutants were generated using *A. tumefaciens*–mediated transformation as described by Ballester et al. [33]. For pathogenicity and virulence tests, one knockout mutant containing a single T-DNA integration was selected for each gene. Transformants with ectopic insertion of transformation cassette, but still possessing the WT gene and the WT strain were used as controls. Notably, the growth rate and colony morphology of *Penlp1* and *Penlp2* on PDA were similar to that of the WT and the ectopic mutants (Appendix A). Spore production as well as germination of the spores was also similar in the null mutants of both NLP genes, relative to the wild type (Appendix A).

Wounded ‘Golden Delicious’ apples were inoculated with spore suspensions of the null and ectopic mutants to examine the effect of the deletion on pathogenicity. The percentage of infected wounds (disease incidence) and the rate of infection development (disease severity) were compared between null mutants, ectopic mutants, and the wild type (WT) strains. Deletion of *Penlp2* gene had no effect on either decay incidence or severity (Figure 5). Inoculation with the Δ*Penlp1* mutant, however, resulted in a significant reduction in lesion dimeter compared to the WT and ectopic mutant (Figure 5). A reduction of 15%, 12% and 9% in lesion diameter was obtained after 4, 5, and 6 days of incubation, respectively. The percentage of infection (disease incidence) in apples inoculated with Δ*Penlp1* was similar to the one observed with the WT and ectopic mutant (Figure 5).

## 4. Discussion

In the current study we wanted to identify NLP genes in *P. expansum*, study their expression during infection and decay development on apple fruit, and determine their potential role in pathogenicity and virulence. We provide the first report of two NLP genes in the *P. expansum* genome. *Penlp1* belongs to the type 1 group and *Penlp2* to the type 3 group. PeNLP1 has the typical NLP conserved heptapeptide motif “GHRHDWE”. This is in agreement with a previous report by Oome and Van den Ackerveken [16] who analyzed 533 NLPs and indicated that all of them have some variant of the conserved heptapeptide motif, “GHRHDWE”. In the case of PeNLP2, the third (Arg) and forth (His) amino acids of this motif are replaced by Asp and Try, respectively. This again is similar to what has been reported for type 3 NLPs.

In general, type 1 NLPs are found in many plant-associated fungal species, whereas type 3 NLPs are only found in Ascomycete fungi with many different lifestyles [16]. The NLP family in oomycetes is significantly expanded. Feng and cowokers [57] identified 42 putative NLPs in the genome of *Phytophthora capsici* containing the conserved “GHRHDWE” motif. The NLP gene family in *P. sojae* is composed of 33 real and 37 pseudogenes [19]. In ascomyceteous fungi, the number of NLP genes is much lower, as was also the case for *P. expansum* in the present study. For example, *B. cinerea* has two NLP genes [31], while one, four and eight NLP genes are present in *Magnaporthe graminicola, M. grisea*, and *V. dahliae*, respectively [14,24,32].

PeNLP1 and PeNLP2 are both cysteine rich with five and six cysteines, respectively. Cysteine residues in NLP proteins form disulfide bonds that are important in protein folding and stability [37,38]. More than 98% of type 1 NLPs contain two conserved cysteine residues in specific locations. These are also present in PeNLP1. Two additional cysteines (Cys153 and Cys162) were also found to be present in PeNLP1 that were adjacent to each other based on the model structure that was generated (Figure 2). Although, the generated model did not predict the formation of a disulfide bond between these cysteine residues, this may be due to the bias of the model. Oome and Van den Ackerveken [16] measured putative distances between carbon-α and carbon-β pairs of various cysteines in type 1 and type 2 NLP sequences available in the databases. In addition to the two conserved cysteines forming the disulfide bridge, more than 100 sequences carrying two or more extra cysteine residues were found. In this study, twelve additional potential disulfide bridges were identified in type 1 and type 2 NLPs. Among these 12 potential disulfide bridges, a bridge equivalent to a CYS153-CYS162 bond was predicted to occur in 43 NLPs. This disulfide bond is found in a branch of fungal type 1 NLP [16], and has been reported to exist in BcNEP2 in *B. cinerea* [30].

The possibility that PeNLP1 and PeNLP2 cause necrosis was assessed using *Agrobacterium* to transiently express the genes in *N. benthamiana* leaves. Agroinfiltration of PeNLP1 to *N. benthamiana* leaves caused necrosis after 8–10 days, while for PeNLP2 took 12 days to cause necrosis symptoms. These results are in agreement with different reports indicating that the time needed to develop necrosis in transient expression systems varies between 3 to 12 days [18,22,23,24,28,57]. Those differences could result from diverse cytotoxic capabilities of various NLPs, the secretion efficiency and/or general experimental conditions. The ability of PeNLP2 to cause necrosis is in line with the claim that amino acids able to form acidic cation-binding pocket necessary for necrosis induction are conserved in type 3 NLPs [16]. Until now, little experimental data available on cytotoxicity of type 3 NLPs, and to our knowledge, the current work report for the first time cytotoxic activity of *P. expansum* type 3 NLP.

The necrosis symptoms caused by transient expression of both PeNLP1 and PeNLP2 with signal peptide was markedly more severe than that caused by the expression of the proteins lacking signal peptide. Signal peptide leads the proteins to the secretion pathway, which supports the proper folding and disulfide bond formation that could be necessary for the NLP function and necrosis development. Similarly, the development of necrosis symptoms induced by transient expression of PsojNIP, type 1 NLP of *P. sojae* without a signal peptide lagged after symptoms developed with the wild-type protein [16]. Type 2 NLPPcc of the bacterium *Pectobacterium carotovorum* as well as four NLPs of *P. capsici* lacking signal peptide did not induce any necrosis [16,23]. Schouten et al. [29], however, showed that BcNEP1 and BcNEP2 transiently expressed in *N. tabacum* leaves with or without endogenous fungal signal peptide were not able to induce necrosis. Only agroinfiltration of constructs in which the cDNAs lacking the signal peptide were cloned in frame with the signal peptide from the tobacco PR1a protein were able to induce necrosis.

Following the discovery of NEP1 in *F. oxysporum* and its ability to cause cell death in dicots [17], many NEP1-like proteins were identified in numerous species of fungi with different lifestyles. Many additional NLPs were described, however, that do not induce necrosis. In *P. capsici*, part of the identified PcNLPs induce chlorosis or necrosis in tobacco or hot pepper leaves [23,57]. Likewise, the NLPs from *B. elliptica*, *B. cinerea,* and *S*. *sclerotiorum* have also been reported to cause necrosis in *N. benthamiana* leaves [28,29,31]. Twelve HaNLPs from *Hyaloperonospora arabidopsidis,* two of the three tested PiNPPs from *P. infestans*, 11 of the 19 tested NLPs in *P. sojae*, seven of the nine NLPs tested in *V. dahliae V592*, six of the eight NLPs tested in *V. dahliae JR2*, and one of the two ChNLPs tested in *C. higginsianum* did not cause necrosis when transiently expressed in leaves [18,19,20,21,22,24]. Notably, no evident correlation between fungal lifestyle or NLP type and cytotoxicity has been identified.

In the present study, *Penlp1* and *Penlp2* were observed to be induced during both decay development and during growth of *P. expansum* in liquid culture. Several reports have investigated the expression of NLP genes in culture and during infection and disease development *in planta*. For example, the expression of *MgNLP* in *M. graminicola*, *VdNLP* in *V. dahlia*, *SSNep1* and *SSNep2* in *S. sclerotiorum*, and *Bcnep1* and *Bcnep2* in *B. cinerea* was most pronounced during infection and disease development [21,24,28,30,32]. Eight NLP-coding genes in *P. capsici* are differentially expressed during plant infection stages [23]. Among them Pc11951 was found to be actively transcribed in plate-grown culture [23]. In our previous work, *PeLysM* effectors and *Peprt* in *P. expansum* were also reported to be actively transcribed in culture, similar to *Penlp1* in the present study [11,12]. These findings may imply a possible involvement of these proteins in fungal growth and development or perhaps that they are induced when the fungus perceives a nutrient-rich environment.

*Penlp1* and *Penlp2* exhibited different expression patterns. *Penlp1* was strongly induced during mycelial growth, while the expression of *Penlp2* was decreased during spore germination and subsequent mycelial growth. Expression of NLP genes in spores during early stages of germination has not been reported in the literature and overall very little information is available about type 3 NLPs, to which *Penlp2* belongs. One could speculate that *Penlp2* expression in the early stages of germination could have a role in modulating fruit resistance mechanisms to facilitate colonization of the infection site and is then followed by the secretion of relatively high levels of PeNLP1 that play a role in modulating disease development. On the other hand, there is possible regulatory role for alternative splicing. As evident in Appendix A, the band representing splice variant encoding complete PeNLP2 protein (var 3) is faint in spores and turns stronger up to 96 hpi. Since other two splice-variants (var 1 and var 2) encode truncated products, the expression of the active form of PeNLP2 is induced during the growth and infection. The *Penlp2* knockout mutant, did not have an effect on disease incidence on apple fruit or fungal growth. This may have been due to the involvement of redundant factors, other than PeNLP2, that play a role in establishing infection and more research will be required to better understand the specific role of the individual NLPs in *P. expansum.*

Understanding fruit-pathogen interactions during the early stages of infection is important for developing control strategies. *P. expansum* must deploy specific pathogenicity and virulence factors to overcome host innate and induced resistance mechanisms. Therefore, to examine the role of NPLs in the pathogenicity and virulence of *P. expansum* on apple fruit, deletion mutants of both genes were created. Results with the knockout mutant strains indicated that only the deletion of PeNLP1 reduced the rate of lesion development (disease severity) but not disease incidence on apples kept at 25 °C (Figure 5). These results are in agreement with earlier studies indicating that a functional diversification exists in the superfamily of NLP1-like proteins with some members contributing to virulence while others do not. For instance, NLP was demonstrated to be essential for the virulence of *Erwinia carotovora* on potato tubers [27]), whereas deletion of VdNLP1 in the wilt pathogen *V. dahliae* JR2 resulted in reduced virulence on tomato, *Arabidopsis,* and *N. benthamiana*. In contrast, deletion of VdNLP2 only affected the level of virulence on tomato and *Arabidopsis* [24]. Deletion of VdNLP1 and VdNLP2 in *V. dahliae* V592 had no effect on symptom development in cotton [21]. Deletion of the only NLP gene from *M. graminicola* did not produce any detectable reduction in pathogenicity or virulence [32]. The loss of NLP genes in *B. cinerea* and *B. elliptica* did not affect virulence either [30,31]. In the present study, *Penlp1* and *Penlp2* deletion mutants exhibited normal spore germination, rate of radial growth, colony morphology, and sporulation on PDA plates (Appendix A). Deletion of NLP1 and NLP2 in *V. dahlia* also had no effect on spore germination and radial growth, however, conidiophore production in NLP1 deletion mutant strains was significantly affected [24].

PCR amplification of *Penlp2* ORF revealed three products (Appendix A). Sequence analysis of these products, revealed three putative products of alternative splicing: var1, var2 and var3. The alternative splicing in *Penlp2* seems to occur via retained introns, which is the dominant form of splice variation in fungi [58]. Analysis of the intron sequences revealed that the intron in var2 is non-canonical. Examination of 2604 introns from 842 genes in *Fusarium graminearum* revealed that 0.31% of the introns are non-canonical and have donor-acceptor sites other than GU-AG or AC-AG [59]. Moreover, the alignment of ESTs with genomic sequences in five diverse fungi revealed that 17% of the sequences from *Schizosaccharomyces pombe*, 11% from *Aspergilus nidulans*, 19% from *Neurospora crassa*, and 7% from *Cryptococcus neoformans* did not conform to the 5′GU…AG3′, 5′GC…AG3′, or 5′AU…AC3′ dinucleotide pairs. Whereas, var3 encodes a predicted gene product (type 3 NLP), there is a stop codon introduced in the reading frame in both var1 and var2. Based on an analysis of alternative splicing events across 42 eukaryotes, including 14 fungi, by McGuire et al. [58], the majority of splice variants lead to frameshifts or truncated proteins. Our results suggest that the expression level and ratio between the three variants may change during vegetative growth. In support of this premise, the regulation of alternative splicing as a function of growth stage was demonstrated for several genes in *F. graminearum* [59]. These findings imply the possibility of a regulatory role for alternative splicing in adaptation to the changing conditions that occur during different stages of growth.

Overall, our results highlights the possible role of NLPs in the pathogenicity of *P. expansum* and would promote our understanding of host-pathogen interactions and possible innovative ways for controlling postharvest decay.

## Figures and Tables

**Figure 1 microorganisms-07-00175-f001:**
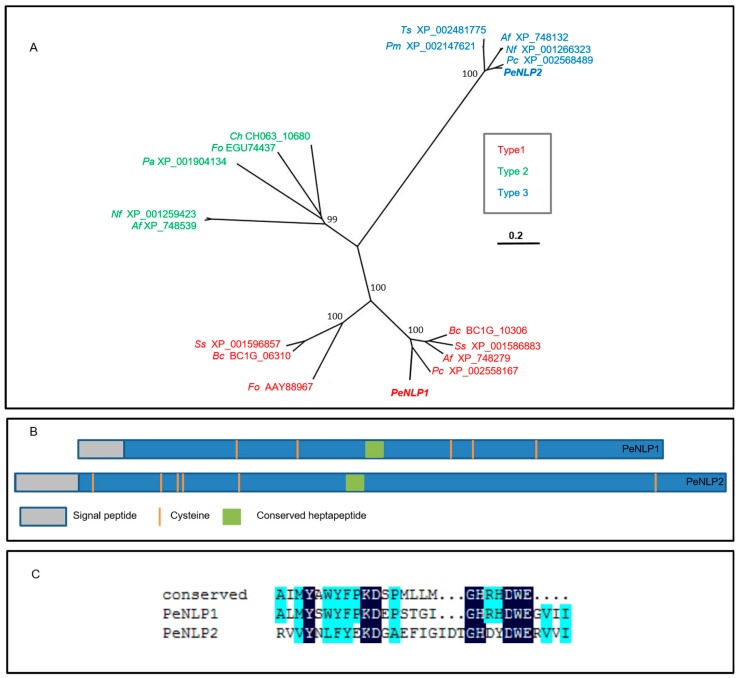
Characterization of the two NLP genes of *P. expansum*. (**A**) An un-rooted phylogenetic tree representing comparison of fungal NLP sequences from GenBank. The alignment was performed using Mega7. NLP type is designated by color. The scale bar represents weighted sequence divergence. The percentage of replicate trees in which the associated taxa clustered together in the bootstrap test (100 replicates) are shown next to the relevant branches. The title of the protein sequences is as follows: organism name is abbreviated by the first two letters, followed by the GenBank protein accession number. Af, *Aspergillus fumigates* Af293; Bc, *B. cinerea*; Ch, *Colletotrichum higginsianum*; Fo, *Fusarium oxysporum*; Nf, *Neosartorya fischeri* NRRL181; Pa, *Podospora anserine* S mat+; Pc, *Penicillium chrysogenum* Wisconsin 54-1255; Pm, *Penicillium marnefferi* ATCC 18224; Ss, *Sclerotinia sclerotiorum* 1980 UF-70, Ts, *Talaromyces stipitatus* ATCC 10500. (**B**) Schematic representation of PeNLP1 and PeNLP2. (**C**) Protein sequence alignment of the conserved nlp24 peptide [25] to the corresponding regions of PeNLP1 and PeNLP2. The amino acids are shaded (100%: black, 50%: light blue).

**Figure 2 microorganisms-07-00175-f002:**
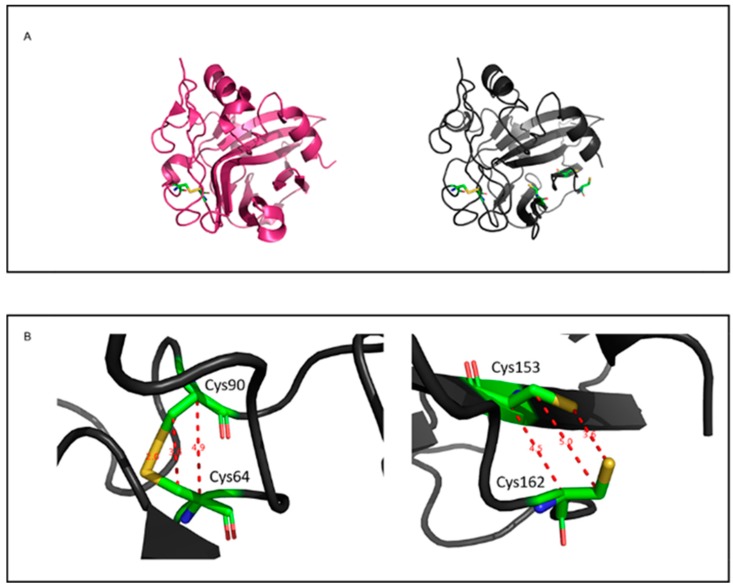
Analysis of potential disulfide bonds in PeNLP1. (**A**) Predicted 3D structures of template (PDB 3ST1) and target (NLP-1 modeled using SWISS-MODEL) in pink and gray, respectively. Cysteine residues are shown in sticks, colored by atoms (carbon in green, oxygen in red, nitrogen in blue and sulfur in yellow). (**B**) Distances between carbon-α, carbon-β and sulfur atoms of cysteine residues in the model structure of NLP-1. The figures were generated using Pymol.

**Figure 3 microorganisms-07-00175-f003:**
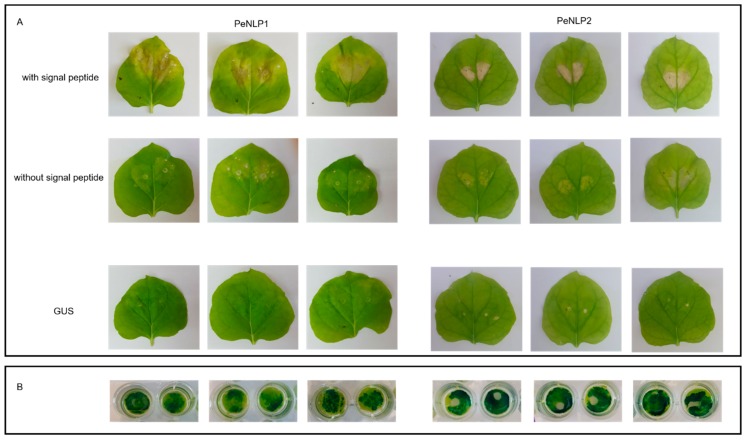
Expression of PeNLP1 and PeNLP2 induces cell death in *N. benthamiana* leaves. (**A**) Leaves of *N. benthamiana* infiltrated with constructs for constitutive expression of PeNLP1 (left panels) and PeNLP2 (right panels) and constutive expression of GUS as negative control. Pictures were taken 10 days post infiltration (**B**) Visualization of GUS activity indicates protein expression in the negative control.

**Figure 4 microorganisms-07-00175-f004:**
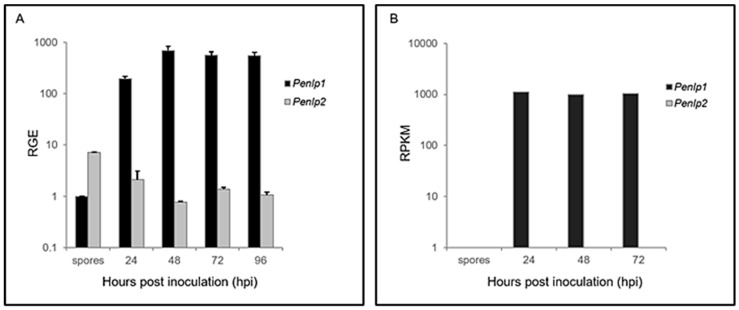
Expression of *Penlp1* and *Penlp2*. (**A**) Relative gene expression of *Penlp1* and *Penlp2*. Apple wounds were inoculated with spore suspensions of *P. expansum* and incubated for 4 days at 25 °C in the dark. Relative gene expression (RGE) in apples wound tissue after 24, 48, 72, and 96 h post inoculation is presented. The expression levels are relative to the genes coding for a 28S rRNA, histone 3, and 37S ribosomal protein s24. Error bars indicate standard errors of three biological replicates. (**B**) Expression level of *Penlp1* and *Penlp2* based on RNA-Seq analysis. RNA-Seq data were generated by Ballester et al. [33] from *P. expansum* (PEX1)–infected apples at 24, 48, and 72 hpi.

**Figure 5 microorganisms-07-00175-f005:**
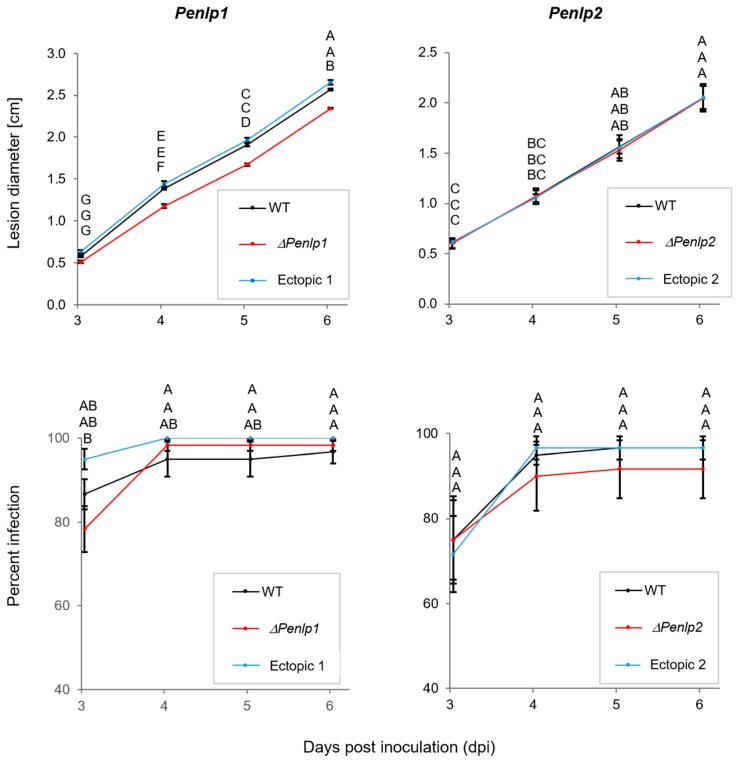
‘Golden Delicious’ apples were inoculated with wild-type and transformant strains to examine pathogenicity. Fruits were wounded and 10 µL of spore suspension (10^5^ conidia/mL) were added into each wound. For each treatment fifteen fruits with four wounds per fruit were used. Decay diameter and percentage of infected wounds were measured at 3, 4, 5, and 6 dpi. Results of one of three independent experiments are presented. Bars indicate standard error. Letters indicate significant differences at *p* < 0.05 based on nested one-way ANOVA followed by Tukey’s honest significant difference (HSD) test.3.5. *Penlp1* and *Penlp2* Knockout Mutants of *P. expansum*.

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
