# Peer review of "Identification and Functional Analysis of NLP-Encoding Genes from the Postharvest Pathogen Penicillium expansum"

_microorganisms, 2019, doi:10.3390/microorganisms7060175_

Round 1

Reviewer 1 Report

The manuscript by Elena Levin and co-author describes two effectors in Penicillium expansum a necrotrophic fungal pathogen that causes post-harvest damages to apples. They describe Penlp1 and Penlp2, two NLP proteins by comparison to other fungal NLPs. They also show the effects of the proteins by infiltration in tobacco, gene expression and knockout mutants.

The manuscript is well written and will be a welcomed piece in the collective description of these fungal NLPs. However, I have minor comments on the description of the results that are lacking in precision.

Line 135: Description of the total branch length is not enough to justify a phylogenetic tree: Although the grouping according to preciously identified types makes this tree stand, the description of the parameters used for the phylogenetic construction, and support for this specific tree is completely lacking. What statistical justification for this particular tree? Significance from bootstrapping could help here.

Line 247: Describe on what exactly is based this “screening of the annotation of the genome”. How did you identify Penlp2? Based on what methods/criteria? Why was it not found by blast?

Line 284 and Figure 2: “The best model structure”. Describe what is “the best”, in comparison to what (what is the worst?), what is the probability of these specific protein models?

Line 308: Please show the construct without peptide signal. Was it more necrotic that the GUS used as negative control? Why is there more necrosis? Show that the difference in necrosis between empty construct and the constructs with PeNLP1 and PeNLP2 is significant.

Figure 3A: what do the circles represent?

I also see that the lesions are different colours and patterns. Quantifying the necrosis as lesion area would be welcomed here.

Figure 3B: I don't understand what is represented here. It’s too small and complete description and why it matters is lacking. Also shortly explain why you are using GUS.

Figure 4: How do you explain that Penlp1 was not induced in the RNA-seq dataset?

Figure 5: Use colour to describe the genotypes instead of symbols. Also, is it average lesion diameter? If yes, also add standard deviation or standard error bars.

Add the description of E1 and E2.

Figure 5B: a percent of infection with an axis going to 120 is awkward. Also zooming on the figure would help see the patterns.

Author Response

On behalf of the authors, I would like to thank reviewer #1 for his comprehensive and constructive review of our manuscript. All comments and suggestion were taken in to consideration and accepted. Explanations to questions raised about specific points were added directly into the text and also included in our response document. We trust managed to provide appropriate answers for all the questions and comments.

The following is our detailed response to all the comments:

Line 135: Description of the total branch length is not enough to justify a phylogenetic tree: Although the grouping according to preciously identified types makes this tree stand, the description of the parameters used for the phylogenetic construction, and support for this specific tree is completely lacking. What statistical justification for this particular tree? Significance from bootstrapping could help here.

Following the comment regarding the justification of the phylogenetic tree and the suggestion to perform bootstrapping, the percentage of replicate trees in which the associated taxa clustered together in the bootstrap test (100 replicates) were calculated and the numbers were added next to the relevant branches of the phylogenetic tree at figure 1. Relevant explanation was added to the materials and method section (line 136).

Line 247: Describe on what exactly is based this “screening of the annotation of the genome”. How did you identify Penlp2? Based on what methods/criteria? Why was it not found by blast?

In our previous work (Levin et al 2015, Postharvest Biology and Technology) we identified a NLP encoding gene (PEX2_080220) with a PF05630.6 (Necrosis inducing protein NPP1) and a IPR008701 (Necrosis inducing protein) domains. We have done a BlastP with the identified NLP protein against the PEX2 proteome and we did not identify any other protein homologue to it. However, the NLP protein contains a NPP domain, so we “screened the annotation of the genome” to find other proteins containing the PF05630.6 or IPR008701 domains, and we identified two proteins: PEX2_080220 and a second one, PEX2_071150. The identity of both proteins is 31.82% along the alignment that covers 38% of the sequence. This is the reason why we did not identify the second protein using BlastP with PEX2_080220 against the PEX2 proteome. It is important to mention that this is in line with the statement of Oome and Ackerveken (2014) according to which: “type 3 NLP share no clear homology with the other two types of NLP, except for the 50 residues surrounding the heptapeptide motif, the rest of the sequence cannot be compared with type 1 and type 2 NLP”

Line 284 and Figure 2: “The best model structure”. Describe what is “the best”, in comparison to what (what is the worst?), what is the probability of these specific protein models?

The prediction of the 3D structure was done using 6 different methods. The quality assessment of each model was done using four independent software packages: Qmean, MolProbity, Verify3D and ProQ2 as described in the Matherials and Methods. Each software evaluates the quality of the model using different algorithm. The scores given by all four programs were taken in to consideration and the model, which got the overall best score was chosen as the “best model”.

Line 308: Please show the construct without peptide signal. Was it more necrotic that the GUS used as negative control? Why is there more necrosis? Show that the difference in necrosis between empty construct and the constructs with PeNLP1 and PeNLP2 is significant.

Pictures of the leaves transiently expressed constructs without signal peptide were added to Figure 3. It can be clearly seen that in the leaves infiltrated with GUS construct, there is no signs for necrosis development, however, in the leaves infiltrated with both PeNLP1 and PeNLP2 without signal peptide, there are necrotic spots and chlorotic area around the infiltration point.

Signal peptide leads the proteins to a secretion pathway, which supports the proper folding and disulfide bond formation that could be necessary for the NLP function and necrosis development. The fact that transient expression of the constructs without signal peptide can still, although with minor extent, cause necrosis could be due to the possibility of obtaining not fully mature NLP protein which still cause very limited necrosis. NLP without signal peptide could be partially folded or folded with lower efficiency, hence causing necrosis with lower efficiency.

The difference between the effect caused by the construct with PeNLP1 and PeNLP2 and the negative control is very clear. While in negative control, no signs of necrosis or other visible damage was observed, there is a distinct necrosis area around each infiltration sight of constructs with PeNLP1 and PeNLP2. The fact that this phenomenon is repetitive makes it significant without a doubt. In Figure 3, we presented 6 biological repeats (6 infiltration sites) in order to emphasize the significance of the effect.

Figure 3A: what do the circles represent?

The circles used to show the infiltration area, but were removed as unnecessary.

I also see that the lesions are different colours and patterns. Quantifying the necrosis as lesion area would be welcomed here.

As rightfully noted by the reviewer, since the lesions are different colors and patterns, quantification of the necrosis as lesion area, could provide more information about the differences in the necrosis efficiency or activity of the two NLP proteins and hence better understanding of they mode of action. However, the scope of this work is initial characterization of the NLP in P. expansum, and the goal of agroinfiltration experiment was to detect whether PeNLP1 and PeNLP2 have cytotoxic activity. Hence, we believe, that the results in the way we present them give a comprehensive answer to this question.

Figure 3B: I don't understand what is represented here. It’s too small and complete description and why it matters is lacking. Also shortly explain why you are using GUS.

In Figure 3B, we present a leave discs taken from the area of agroinfiltration of the negative control (construct expressing GUS). The blue color in the discs is a result of GUS activity. Visualization of GUS activity indicates that there is an expression of the protein in the negative control and stress the fact that necrotic activity is specific to NLPs and not an artifact of nonspecific ectopic protein expression. Relevant explanation was added to the caption of Figure 3.

GUS was chosen as a negative control, since it does not have any cytotoxic activity and its expression can be easily detected and visualized (no need of Western blot). Due to those reasons GUS is widely used as a control in similar experiments.

Figure 4: How do you explain that Penlp1 was not induced in the RNA-seq dataset?

As we can observe in Figure 4B, in the RNA-seq experiment, the expression of Penlp1 (PEX2_080220) was not detected in spores, but after 24 hours post inoculation there is an induction of the expression of the gene, with a RPKM value of 1100, that was maintained at 48 and 72 hpi. Therefore, there is an important induction of Penlp1 at 24 hpi compared to spores. Similar results were observed by RTqPCR in other different experiment with other samples (Figure 4A). In this experiment, the relative expression of Penlp1 gene in spores was 1, and there was an 100-fold induction after 24 hpi.

Figure 5: Use color to describe the genotypes instead of symbols. Also, is it average lesion diameter? If yes, also add standard deviation or standard error bars.

The description of the genotypes in Figure 5 was changed to color as suggested. The lesion diameter represented in the graphs are average (as described in the caption) and the relevant standard errors are presented.

Add the description of E1 and E2.

E1 and E2 stand for ectopic mutants of PeNLP1 and PeNLP2 respectively. The description in the figure was changed accordingly.

Figure 5B: a percent of infection with an axis going to 120 is awkward. Also zooming on the figure would help see the patterns.

The axis in the relevant graphs (Figure 5B) was changed: the maximum was set to 100%, and the minimum to 40 % to enable zooming.

Reviewer 2 Report

The authors presented the identification and functional analysis of NLP-encoding genes from Penicillium expansum. The content of this manuscript is well organized. This manuscript contains content that is of interest to experts in this field as well as non-experts. If possible, the authors should include comments, mainly for non-experts, on the future application of the results obtained in this study to relevant fields.

Author Response

We thank reviewer #2 for his positive comments and insights. His comment regarding the importance of NLP-encoding genes is important and relevant explanation was added to the text (line 516).

This work highlights that it the first time that NLPs were found in the necrotrophic pathogen, P. expansum, and both proteins are shown to have cytotoxic effects in plant tissue. Moreover, the possible involvement of PeNLP1 in pathogenicity would suggest a similar role of this effector in other plant-pathogen systems. Understanding the involvement specific effectors in pathogenicity of fungal plant pathogens would facilitate in the future development of innovative control means that are targeted to specific sites.